# Stability Characteristics of Horizontal Wells in the Exploitation of Hydrate-Bearing Clayey-Silt Sediments

Xiaofeng Sun [1], Qiaobo Hu [1,2,3,*], Yanlong Li [2,3,*], Mingtao Chen [2,3,4] and Yajuan Zhang [2,3]

1   Sanya Offshore Oil and Gas Research Institute, Northeast Petroleum University, Sanya 572025, China
2   Key Laboratory of Gas Hydrate, Ministry of Natural Resources, Qingdao Institute of Marine Geology, Qingdao 266237, China
3   Laboratory for Marine Mineral Resources, Pilot National Laboratory for Marine Science and Technology, Qingdao 266237, China
4   College of Oceanography, Hohai University, Nanjing 210098, China
*   Correspondence: hqbnepu@stu.nepu.cn (Q.H.); ylli@qnlm.ac (Y.L.)

**Abstract:** The mechanical properties of hydrate-bearing strata in clayey-silt sediments are significantly different from those of either conventional reservoirs or hydrate-bearing sandy sediments, which poses great challenges for wellbore stability analyses. The stability characteristics of a deviated borehole during drilling in hydrate-bearing clayey-silt sediments (HBS-CS) remain to be studied. In this paper, an analysis of the wellbore stability characteristics of a deviated borehole using the Mohr–Coulomb (M-C) criterion and Drucker–Prager (D-P) criterion was carried out based on the elastic stress distribution model of the surrounding strata of the wellbore and the triaxial shear tests of the HBS-CS. The results imply that the collapse pressure and safety density window are symmetrically distributed with deviation angle and azimuth. Considering the effect of hydrate decomposition, the collapse pressure gradient could become higher and the instability risks would be amplified. Considering the combined effects of collapse, fracture pressure gradient, and the safety density window, it is suggested that the borehole be arranged along an azimuth of 60–120°, which could greatly reduce the risk in a drilling operation.

**Keywords:** natural gas hydrate; clayey-silt; deviated borehole; stability characteristics; hydrate saturation; collapse pressure; fracture pressure; safety density window



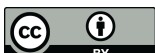

## 1. Introduction

Natural gas hydrate (NGH) is a kind of low-carbon fossil energy, and vigorously developing NGH is of great significance [1–3]. At present, more than 30 countries and/or regions have been actively involved in the development of NGH. In 1976, Russia began to exploit NGH from the permafrost of the Messoyakha [4,5]. In 2002, Japex jointly carried out the NGH pilot production in the Mackenzie delta of Canada in 2002 and 2007–2008 [6,7]. In 2008, the United States successfully exploited NGH on the northern slopes of Alaska by using $CO_2$ replacement [8,9]. In 2013, Japan completed the world's first offshore NGH development in the Nankai Trough [10,11]. In 2017 [12] and 2020 [13], China completed two rounds of NGH production trials from the clayey-silt sediment in the northern South China Sea.

To improve the gas productivity from HBS-CS, China's second marine NGH production trial was carried out via a horizontal wellbore. In the process of wellbore construction, the whipstock section passes through the overlaying strata and extends horizontally in the NGH-bearing layer. One of the most crucial challenges during horizontal well drilling is to control the stability of the borehole [14–16], which is highly coupled with fluid circulation and fluid losses [17]. For the study of NGH production wellbore stability, Freij-Ayoub et al. (2016) established a wellbore stability analytic model for NGH production [18]. Zhang et al. (2018) analyzed the effect of temperature on the stability of the NGH

production borehole [19]. Sun et al. (2018) analyzed the effects of drilling fluid, and the initial reservoir conditions based on the geological backgrounds of the first exploration in the Shenhu area, northern South China Sea [20]. Yuan et al. (2020) simulated the wellbore stability of a vertical wellbore during depressurization [21]. These studies mainly focus on the wellbore stability of vertical or horizontal wellbores. However, the stability of the wellbore of highly deviated sections was rarely reported. Additionally, the M-C criterion was mostly adopted in the failure evaluation, but the influence of intermediate principal stress was ignored, which might affect the calculation accuracy of wellbore stability.

In this paper, the stability characteristics of an inclined wellbore during drilling are analyzed. The failure criteria of the M-C and the D-P are deployed, respectively. Taking the collapse pressure, fracture pressure, and the safety density window as the main evaluation indices, the effects of hydrate saturation ($S_h$), the deviation angle, and the azimuth on the stability of the borehole are clarified. The results may have some significance for drilling design in the clayey-silt HBS in the northern South China Sea.

## 2. Mechanical Parameters of HBS-CS

The basic mechanical parameters of the HBS-CS were obtained from triaxial shear tests. A detailed description of the experimental devices can be found in our previous publication [22,23]. The steps of the triaxial shear tests are as follows: (1) HBS-CS skeleton prefabrication; (2) triaxial testing system installation; (3) apply confining pressure; and (4) hydrate formation, which has real-time monitoring of the hydrate saturation by time domain reflectometry, and finally the triaxial shear tests.

The sediment in the Shenhu area, northern South China Sea is mainly composed of argillaceous siltstone with poor cementation strength and a high content of silt and clay [23,24]. The particle size distribution of the sediment is shown in Figure 1, in which the medium grain size is about 7 μm. The actual $S_h$ range is between 20% and 60% [14], and hence, the predetermined $S_h$ is 15%, 30%, 45%, and 60% in the experiments. The effective confining pressure was set as 1 MPa, 2 MPa, and 4 MPa in the triaxial shear tests, respectively.

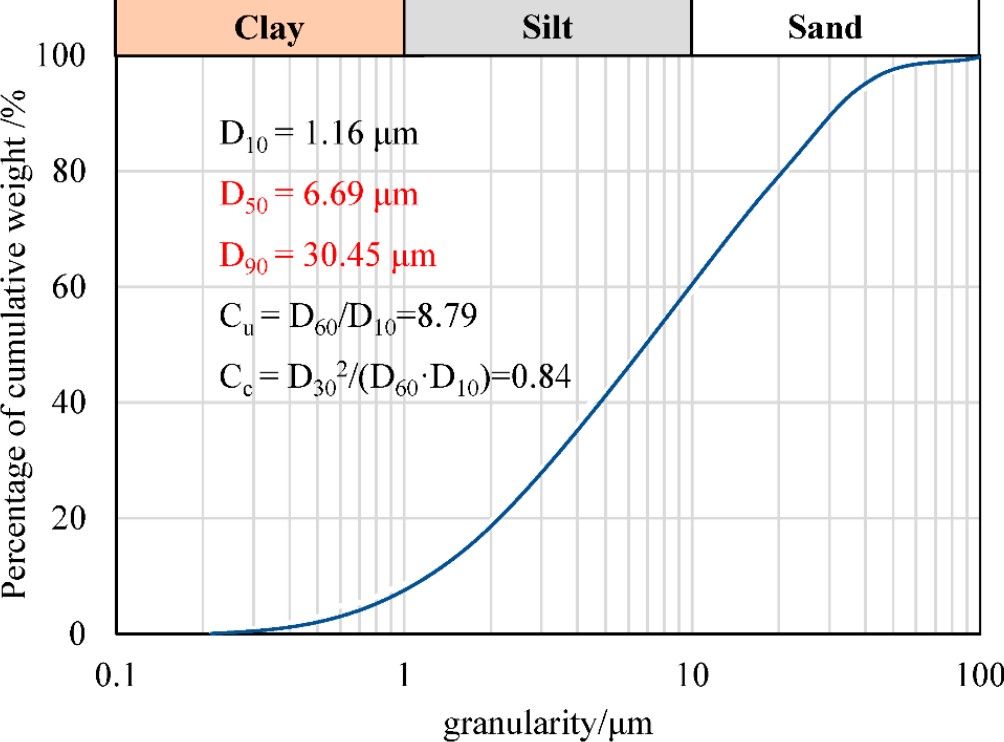

**Figure 1.** Particle size distribution of the filler.

The basic mechanical parameters of HBS-CS were obtained, as shown in Table 1. It could be seen that the cohesion and the internal friction angle of the HBS-CS are much lower than that of the conventional natural gas reservoirs (Li et al., 2020), and the cohesion and friction angle increased with the increase in $S_h$. For a detailed analysis of mechanical properties, refer to the literature [25,26].

**Table 1.** Mechanical parameters of the hydrate.

| $S_h$ | $\sigma_3$/MPa | $\sigma_1$/MPa | Cohesion C/MPa | Friction Angle $\phi$/° |
|---|---|---|---|---|
| 15% | 1 | 1.62 | 0.0953 | 10.340 |
|  | 2 | 3.17 |  |  |
|  | 4 | 5.95 |  |  |
| 30% | 1 | 2.48 | 0.426 | 11.612 |
|  | 2 | 4.16 |  |  |
|  | 4 | 7.03 |  |  |
| 45% | 1 | 3.54 | 0.811 | 12.275 |
|  | 2 | 5.12 |  |  |
|  | 4 | 8.17 |  |  |
| 60% | 1 | 4.29 | 1.071 | 13.568 |
|  | 2 | 6.00 |  |  |
|  | 4 | 9.15 |  |  |

## 3. Model Description

### 3.1. Stress Distribution around the Wellbore

Currently, scholars have mainly investigated borehole stability in NGH-bearing sediments by referring to similar problems in conventional formations. NGH-bearing sediments are always regarded as elastic or elastic–plastic materials when doing borehole stability analysis. In the analysis of collapse pressure, the elastic model was often used [18,20,27,28]. Therefore, in this study, the HBS-CS formation is assumed to be an elastic material, and an elastic model is established to analyze the wellbore stability. The elastic model has few parameters and clear physical meaning. The calculation results of the model are analytical solutions rather than numerical solutions, and the calculation speed is extremely fast. It can be used for the rapid analysis of the situation in a well. It has great advantages in dealing with emergencies.

3.1.1. Conversion of Axis Coordinates of Wellbore

In the process of calculating the stress surrounding a deviated wellbore, the coordinate transformation should be carried out first. The principal in situ stress coordinate ($x_2$, $y_2$, $z_2$) shall be converted into a wellbore axis coordinate system ($x$, $y$, $z$). The in situ stress coordinate system, corresponding to the $Ox_2$ axis, $Oy_2$ axis, and $Oz_2$ axis is consistent with the principal ground stresses $\sigma_H$, $\sigma_h$, and $\sigma_v$ directions, respectively. In the wellbore axis coordinate system, axis $Oz$ corresponds to the deviated wellbore axis and axes $Ox$ and $Oy$ are located in a plane perpendicular to the well axis, and $\psi$, and $\varphi$ are the deviation angle and azimuth, respectively (Figure 2).

The principal in situ stress coordinate system rotates with the wellbore axis, and the coordinate system transformation relationship can be derived from direction cosine, $M_{(\varphi,\psi)}$, for which the expressions are listed as follows:

$$M_{(\varphi)} = \begin{pmatrix} \cos\varphi & -\sin\varphi & 0 \\ \sin\varphi & \cos\varphi & 0 \\ 0 & 0 & 1 \end{pmatrix} \quad M_{(\psi)} = \begin{pmatrix} \cos\psi & 0 & \sin\psi \\ 0 & 1 & 0 \\ -\sin\psi & 0 & \cos\psi \end{pmatrix} \quad (1)$$

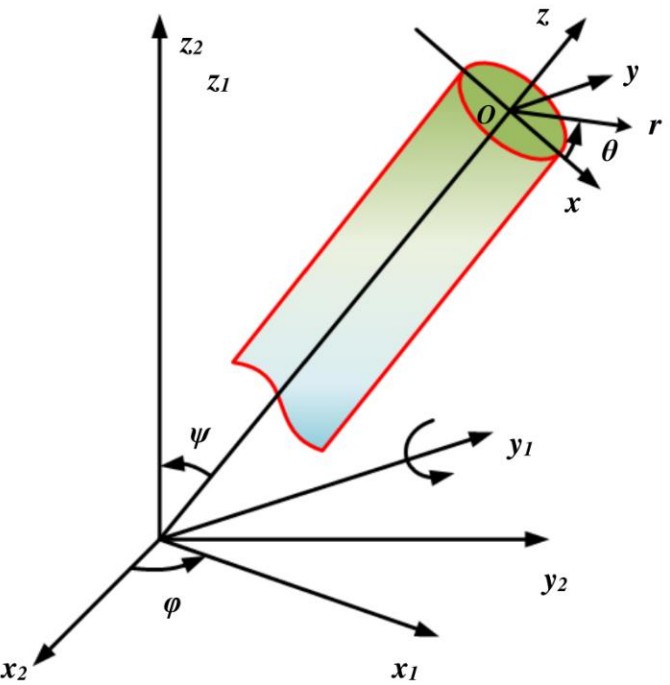

**Figure 2.** Coordinate transformation diagram of a deviated well axis.

$$M_{(\varphi,\psi)} = M_{(\varphi)} \cdot M_{(\psi)} = \begin{pmatrix} \cos\varphi\cos\psi & -\sin\varphi & \cos\varphi\sin\psi \\ \sin\varphi\cos\psi & \cos\varphi & \sin\varphi\sin\psi \\ -\sin\psi & 0 & \cos\psi \end{pmatrix} \quad (2)$$

where $\varphi$ is the deviation angle of the wellbore in °; $\psi$ is the azimuth in °.

In the coordinate system $(x, y, z)$, each stress components calculation formula is as follows:

$$\begin{pmatrix} \sigma_{xx} & \tau_{xy} & \tau_{xz} \\ \tau_{yx} & \sigma_{yy} & \tau_{yz} \\ \tau_{zx} & \tau_{zy} & \sigma_{zz} \end{pmatrix} = M_{(\varphi,\psi)} \cdot \begin{pmatrix} \sigma_H & 0 & 0 \\ 0 & \sigma_h & 0 \\ 0 & 0 & \sigma_v \end{pmatrix} \cdot M_{(\varphi,\psi)}^T \quad (3)$$

where $\sigma_H$, $\sigma_h$, and $\sigma_v$ represent the in situ stresses in MPa; $\sigma_{xx}$, $\sigma_{yy}$, and $\sigma_{zz}$ are the principal stresses in MPa in directions, $x$, $y$, and $z$, respectively; and $\tau$ is the shear stress in MPa.

Since the wellbore section is circular, polar coordinates can be used for simplicity. The cartesian coordinates to polar coordinates transformation is performed as follows:

$$P_{(\theta)} = \begin{pmatrix} \cos\theta & -\sin\theta & 0 \\ \sin\theta & \cos\theta & 0 \\ 0 & 0 & 1 \end{pmatrix} \quad (4)$$

where $\theta$ is the wellbore rounded angle in °.

Referring to Equation (3), transform to polar coordinates $(r, \theta, z)$, and each stress component can be expressed as:

$$\begin{cases} \sigma_r = \sigma_{xx}\cos^2\theta + \sigma_{yy}\sin^2\theta + 2\tau_{xy}\sin\theta\cos\theta \\ \sigma_\theta = \sigma_{xx}\sin^2\theta + \sigma_{yy}\cos^2\theta - 2\tau_{xy}\sin\theta\cos\theta \\ \sigma_z = \sigma_{zz} \\ \tau_{\theta r} = (\sigma_{yy} - \sigma_{xx})\sin\theta\cos\theta + \tau_{xy}(\cos^2\theta - \sin^2\theta) \\ \tau_{zr} = \tau_{xz}\cos\theta + \tau_{yz}\sin\theta \\ \tau_{z\theta} = -\tau_{xz}\sin\theta + \tau_{yz}\cos\theta \end{cases} \quad (5)$$

where $\sigma_r$, $\sigma_\theta$, and $\sigma_z$ are the radial stresses of the reservoir, the tangential stress of the reservoir, and the vertical stress of the reservoir, respectively, in MPa.

### 3.1.2. The Elastic Solution of the Stress around the Wellbore

In polar coordinates ($r$, $\theta$, $z$), the problem of the redistribution of stresses generated by the principal in situ stress $\sigma_H$, $\sigma_h$, and $\sigma_v$ on the surrounding strata of the wellbore is analyzed. It is assumed that the surrounding strata of the wellbore located in the HBS-CS are homogeneous, isotropic, linear–elastic, without creep, and independent from viscosity behavior. The stress of the original strata is in an isotropic state. The analytical solution is solved according to the principle of elasticity. The stress on the surrounding strata meets the plane-stress mechanics' equilibrium and consistent equation [29], in which the plane-stress equilibrium equation can be expressed as:

$$\begin{cases} \frac{\partial \sigma_r}{\partial r} + \frac{\partial \tau_{\theta r}}{r \partial \theta} + \frac{\sigma_r - \sigma_\theta}{r} = 0 \\ \frac{\partial \tau_{\theta r}}{\partial r} + \frac{\partial \sigma_\theta}{r \partial \theta} + \frac{2\tau_{\theta r}}{r} = 0 \end{cases} \tag{6}$$

The consistent equation is expressed as:

$$\left( \frac{\partial^2}{\partial r^2} + \frac{\partial}{r \partial r} + \frac{\partial^2}{r^2 \partial \theta^2} \right) (\sigma_r + \sigma_\theta) = 0 \tag{7}$$

It is assumed that the pore pressure is independent of time and the borehole radius is at any position of the borehole. According to Equation (5), the stress components and boundary conditions could be obtained by the independent action of liquid column pressure, $p_i$, and principal stress, $\sigma_{xx}$, $\sigma_{yy}$, $\tau_{xy}$, $\tau_{xz}$, and $\tau_{yz}$. Equation (5) is substituted into Equations (6) and (7) to obtain the surrounding strata stresses of the wellbore (Tables 2 and 3). In the vertical well, each component in Table 3 is 0.

**Table 2.** Stress distribution expression for the surrounding strata caused by $p_i$, $\sigma_{xx}$, $\sigma_{yy}$, $\tau_{xy}$, $\tau_{xz}$, and $\tau_{yz}$.

| $\sigma_{\mathbf{r}}$ | $\sigma_{\boldsymbol{\theta}}$ | $\tau_{\boldsymbol{\theta}\mathbf{r}}$ |
|---|---|---|
| $\frac{R^2}{r^2} p_i$ | $-\frac{R^2}{r^2} p_i$ | / |
| $\left(1 - \frac{R^2}{r^2}\right)\frac{\sigma_{xx}}{2} + \left(1 - \frac{4R^2}{r^2} + \frac{3R^4}{r^4}\right)\frac{\sigma_{xx}}{2}\cos 2\theta$ | $\left(1 + \frac{R^2}{r^2}\right)\frac{\sigma_{xx}}{2} - \left(1 + \frac{3R^4}{r^4}\right)\frac{\sigma_{xx}}{2}\cos 2\theta$ | $-\left(1 + \frac{2R^2}{r^2} - \frac{3R^4}{r^4}\right)\frac{\sigma_{xx}}{2}\sin 2\theta$ |
| $\left(1 - \frac{R^2}{r^2}\right)\frac{\sigma_{yy}}{2} - \left(1 - \frac{4R^2}{r^2} + \frac{3R^4}{r^4}\right)\frac{\sigma_{yy}}{2}\cos 2\theta$ | $\left(1 + \frac{R^2}{r^2}\right)\frac{\sigma_{yy}}{2} + \left(1 + \frac{3R^4}{r^4}\right)\frac{\sigma_{yy}}{2}\cos 2\theta$ | $\left(1 + \frac{2R^2}{r^2} - \frac{3R^4}{r^4}\right)\frac{\sigma_{yy}}{2}\sin 2\theta$ |
| $\left(1 - \frac{4R^2}{r^2} + \frac{3R^4}{r^4}\right)\tau_{xy}\sin 2\theta$ | $-\left(1 + \frac{3R^4}{r^4}\right)\tau_{xy}\sin 2\theta$ | $\left(1 + \frac{2R^2}{r^2} - \frac{3R^4}{r^4}\right)\tau_{xy}\cos 2\theta$ |

**Table 3.** Stress distribution expression for the surrounding strata caused by $\tau_{xy}$, and $\tau_{yz}$.

| $\tau_{\mathbf{rz}}$ | $\tau_{\boldsymbol{\theta}\mathbf{z}}$ |
|---|---|
| $\tau_{xz}\left(1 - \frac{R^2}{r^2}\right)\cos\theta$ | $-\tau_{xz}\left(1 + \frac{R^2}{r^2}\right)\sin\theta$ |
| $\tau_{yz}\left(1 - \frac{R^2}{r^2}\right)\sin\theta$ | $\tau_{yz}\left(1 + \frac{R^2}{r^2}\right)\cos\theta$ |

The stresses of the surrounding strata can be solved according to Hooke's law. The expression of Hooke's law in direction $z$ is as follows:

$$\varepsilon_z = \frac{1}{E}\left[\sigma_{zz} - v\left(\sigma_{xx} + \sigma_{yy}\right)\right] \tag{8}$$

where $\varepsilon_z$ and $v$ are the strain in direction $z$, and the Poisson's ratio, respectively, and are dimensionless; $E$ is the elastic modulus in MPa.

Neglecting the deformation of the formation in the vertical direction $z$, and substituting $\varepsilon_z = 0$ into Equation (8), the equation is transformed into:

$$\sigma_{zz} = v\left(\sigma_{xx} + \sigma_{yy}\right) \tag{9}$$

Following this step, the component stresses of directions $r$ and $\theta$ (Table 2) are summed to obtain the expression of the wellbore surrounding stress caused by the overburden pressure.

$$\sigma_z = \sigma_{zz} - v\left[2(\sigma_{xx} - \sigma_{yy})\frac{R^2}{r^2}\cos 2\theta + 4\tau_{xy}\frac{R^2}{r^2}\sin 2\theta\right] \tag{10}$$

The linear superposition of all the stress components obtains the stress component at the shaft wall ($r = R$). The principal in situ stresses $\sigma_H$, $\sigma_h$, and $\sigma_v$ are obtained by substituting the stress component of the wellbore axis coordinate system. These expressions are as follows:

$$\begin{cases} \sigma_r = p_i \\ \sigma_\theta = A\sigma_H + B\sigma_h + C\sigma_v - p_i \\ \sigma_z = D\sigma_H + E\sigma_h + F\sigma_v \\ \sigma_{\theta z} = G\sigma_H + H\sigma_h + J\sigma_v \\ \sigma_{r\theta} = \sigma_{rz} = 0 \end{cases} \tag{11}$$

where

$$\begin{cases} A = \cos\psi\left[\cos\psi\cos^2\varphi(1 - 2\cos 2\theta) - 2\sin 2\varphi\sin 2\theta\right] + \cos^2\varphi(1 + 2\cos 2\theta) \\ B = \cos\psi\left[\cos\psi\sin^2\varphi(1 - 2\cos 2\theta) + 2\sin 2\varphi\sin 2\theta\right] + \cos^2\varphi(1 + 2\cos 2\theta) \\ C = \sin^2\psi(1 - 2\cos 2\theta) \\ D = \sin^2\psi\cos^2\varphi - 2v\left[\cos\psi\sin 2\varphi\sin 2\theta - (\sin^2\varphi - \cos^2\psi\cos^2\varphi)\cos 2\theta\right] \\ E = \sin^2\psi\sin^2\varphi + 2v\left[\cos\psi\sin 2\varphi\sin 2\theta + (\cos^2\varphi - \cos^2\psi\sin^2\varphi)\cos 2\theta\right] \\ F = \cos^2\psi - 2v\sin^2\psi\cos 2\theta \\ G = \sin\psi\sin 2\varphi\cos\theta - \sin 2\psi\cos^2\varphi\sin\theta \\ H = -\left(\sin\psi\sin 2\varphi\cos\theta + \sin 2\psi\sin^2\varphi\sin\theta\right) \\ J = \sin 2\psi\sin\theta \end{cases} \tag{12}$$

The rock unit on the deviated wellbore is shown in Figure 3. Since $\sigma_r$ is the principal stress, the deviated shaft wall is still a principal stress surface. To judge the location of a rock failure, the other two principal stress planes must be solved first.

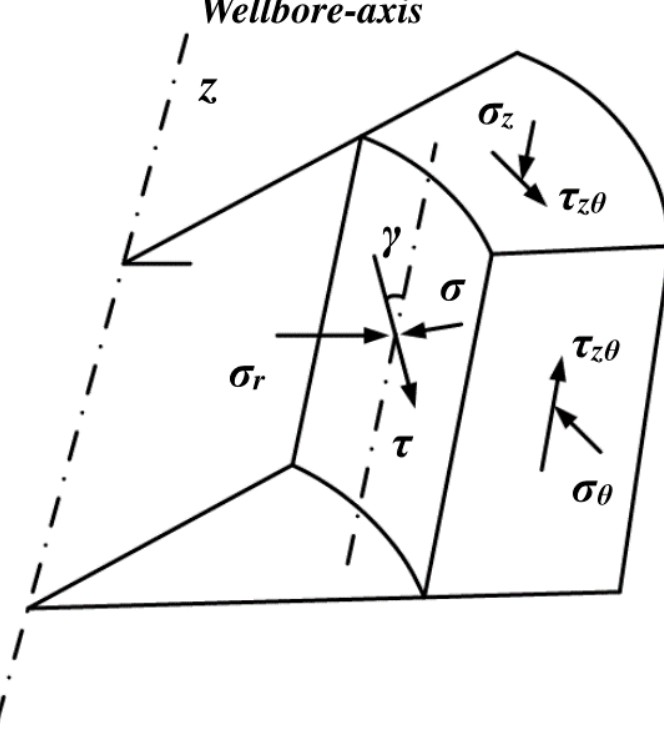

**Figure 3.** Schematic diagram of the stress distribution of the rock element on a deviated wellbore.

According to stress analysis, the relationship between the normal stress, $\sigma_r$, shear stress, $\tau$, and each component is as follows:

$$\begin{cases} \sigma = \sigma_\theta \cos^2\gamma + \sigma_z \sin^2\gamma + 2\tau_{\theta z}\sin\gamma\cos\gamma \\ \tau = \frac{1}{2}(\sigma_z - \sigma_\theta)\sin 2\gamma + \tau_{\theta z}\cos 2\gamma \end{cases} \tag{13}$$

by substituting $d\sigma/d\gamma = 0$ into Equation (13), the two principal stress expressions can be obtained. Considering the influence of the pore pressure, the expression of the stress distribution of the hydrate-deviated wellbore can be obtained as follows:

$$\begin{cases} \sigma_i = \sigma_r = p_i - \alpha p_p \\ \sigma_j = \frac{1}{2}(\sigma_\theta + \sigma_z) + \frac{1}{2}\sqrt{(\sigma_\theta - \sigma_z)^2 + 4\tau_{\theta z}{}^2} - \alpha p_p \\ \sigma_k = \frac{1}{2}(\sigma_\theta + \sigma_z) - \frac{1}{2}\sqrt{(\sigma_\theta - \sigma_z)^2 + 4\tau_{\theta z}{}^2} - \alpha p_p \end{cases} \tag{14}$$

where $\sigma_\theta$ and $\sigma_z$ are the expressions as in Equation (11); $p_p$ is the pore pressure in MPa; $\alpha$ is the effective stress coefficient and is dimensionless.

### 3.2. Failure Criteria

In this study, two failure criteria were used, namely the M-C failure criterion and the D-P failure criterion [30,31]. The M-C failure criterion is the most widely used criterion when analyzing the stability of the boreholes during drilling [32,33]. The criterion, however, only considers the effect of maximum and minimum principal stresses and ignores the influence of the intermediate principal stress. On the other hand, the D-P failure criterion also takes into account the influence of the intermediate principal stress, and adds hydrostatic pressure, thereby overcoming the main weakness of the M-C failure criterion. The D-P failure criterion was also applied in the numerical analysis of formation stability at home and abroad.

#### 3.2.1. The M-C Failure Criterion

Coulomb proposed that the failure of rock was mainly caused by shear failure. The strength of the rock, namely the frictional strength, was supposed to be equal to the adhesion force of the rock, itself against the friction imposed by the shear, and the friction force generated by the normal force on the shear surface. The general form is as follows [34]:

$$\tau = C + \sigma\tan\phi \tag{15}$$

where $\tau$, $\sigma$, and $C$, all in MPa, are the shear stress, the normal stress on the shear plane, and the cohesive force, respectively; $\phi$ is the angle of internal friction in $^\circ$.

To facilitate the calculation, the principal stress form is rewritten as:

$$\sigma_1 = \sigma_3 K^2 + 2CK \tag{16}$$

where $K^2$ is the influence coefficient of confining pressure on axial bearing capacity, $K = \cot\left(\frac{\pi}{4} - \frac{\phi}{2}\right)$.

#### 3.2.2. The D-P Failure Criterion

The D-P failure criterion is an extension of the M-C failure criterion and the von Mises failure criterion [35]. The difference from the M-C failure criterion is that the corner of the yield surface for the D-P failure criterion is smooth and conical in the principal stress space, eliminating the singular point caused by sharp angles, thereby facilitating numerical calculation of the stability of the wellbore (Figure 4).

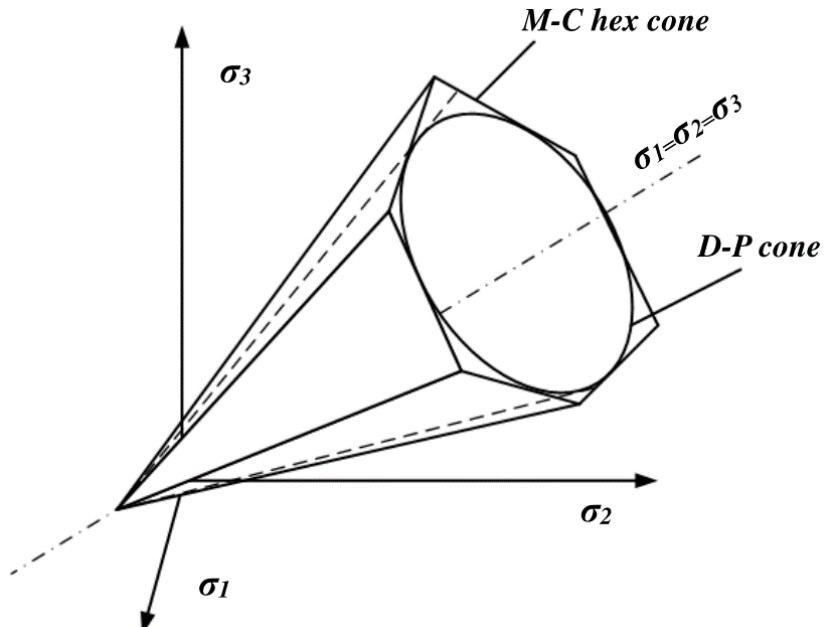

**Figure 4.** A comparison between the three-dimensional failure surface derived from the M-C criterion and the D-P criterion.

The standard form of the D-P failure criterion is as follows [36]:

$$f = Q_f I_1(\sigma_{ij}) + \sqrt{J_2(S_{ij})} + K_f = 0 \tag{17}$$

where $f$, $I_1(\sigma_{ij})$, and $J_2(S_{ij})$ are the plastic potential function, the first invariant of the stress tensor, and the second invariant of the stress partial tensor, respectively. $Q_f$ and $K_f$ are a function of cohesion $C$, and internal friction angle $\phi$, respectively, as follows:
$Q_f = \frac{\sqrt{3}\sin\phi}{3\sqrt{3+\sin^2\phi}}, K_f = -\frac{\sqrt{3}C\cos\phi}{\sqrt{3+\sin^2\phi}}.$

*3.3. Collapse Pressure and Fracture Pressure*

Since the M-C failure criterion only considers the influence of the maximum and the minimum principal stresses and ignores the influence of the intermediate principal stress, it is necessary to establish the principal stress Equation (14) to obtain the maximum and the minimum principal stresses. Then, by substituting them into the M-C failure criterion Equation (16), the value of $p_i$ that makes the equation hold was obtained, which is the collapse pressure of the wellbore under the M-C failure criterion.

Similarly, by substituting these three principal stresses into the D-P failure criterion Equation (17), the collapse pressure of the hydrate wellbore can be obtained by the D-P failure criterion.

For the calculation of the fracture pressure, since only $\sigma_k$ can be negative values in Equation (16), the fracture pressure calculation expression is as follows:

$$f(p_b) = \frac{1}{2}(\sigma_\theta + \sigma_z) - \frac{1}{2}\sqrt{(\sigma_\theta - \sigma_z)^2 + 4\tau_{\theta z}^2} - \alpha p_p + S_t = 0 \tag{18}$$

where $S_t$ is the uniaxial tensile strength of the HBS-CS in MPa.

The above equations were substituted into Matlab to solve for the collapse and fracture pressure of the HBS-CS-deviated wellbore.

## 4. Wellbore Stability Analyses

The basic formation parameters are shown in Table 4. The collapse and fracture pressure of the deviated wellbore located in the HBS-CS are obtained.

**Table 4.** Basic parameter values [37,38].

| Parameter | Value |
|---|---|
| Maximum principal stress, g/cm$^3$ | 1.203 |
| Minimum principal stress, g/cm$^3$ | 1.159 |
| Overburden rock stress, g/cm$^3$ | 1.282 |
| Poisson's ratio | 0.45 |
| Effective stress factor | 0.6 |

*4.1. Effect of $S_h$ on Collapse Pressure of Wellbore*

To describe the stability characteristics of the deviated wellbore located in HBS-CS, the lower hemisphere projection method proposed by Zoback was adopted [39] to show the collapse pressure gradient distribution cloud diagram with different saturation at different azimuth and deviation angles. The cloud diagram of the collapse pressure gradient distribution around the deviated wellbore with different saturation by the M-C and D-P failure criteria are shown in Figures 5 and 6, respectively.

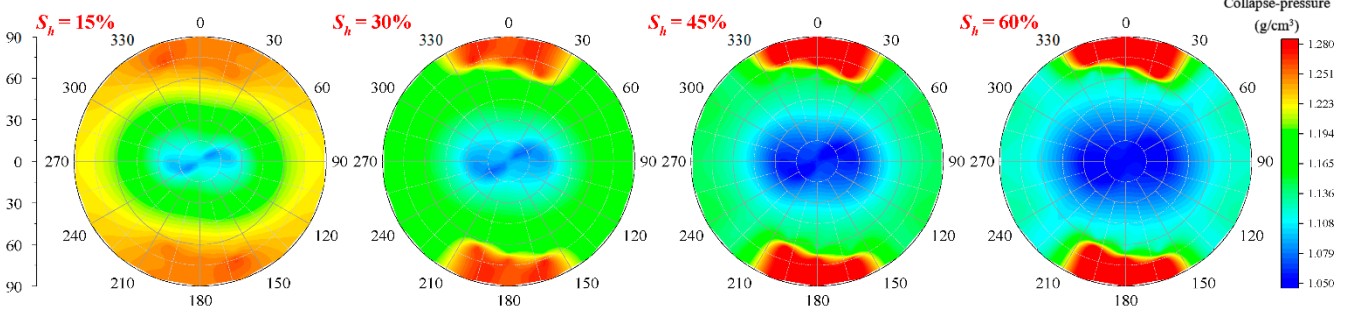

**Figure 5.** Cloud diagram of the collapse pressure gradient distribution under M-C failure criterion.

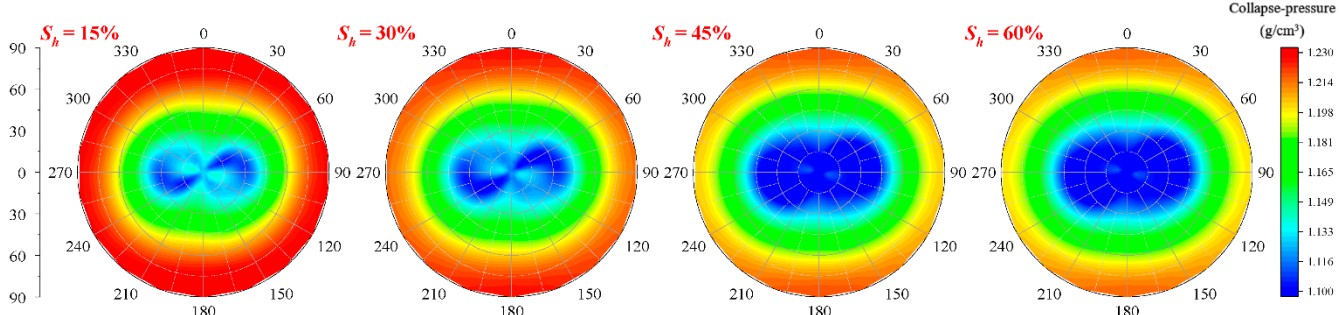

**Figure 6.** Cloud diagrams for the collapse pressure gradient distribution under the D-P failure criterion.

A brief comparison between Figures 5 and 6 implies that the collapse pressure gradient varies with the azimuth in a centrosymmetric distribution at different saturations. This is independent of the chosen failure criteria. The low collapse pressure gradient area is concentrated in the middle "8" region. However, there are differences in the distribution locations of the high collapse pressure gradient area for the different criteria. In the M-C failure criterion, the area where the azimuth ranges over 330°-30° and the deviation angle is greater than 75° has the largest collapse pressure gradient. On the other hand, the collapse pressure gradient is the highest in the annular region (the maximum principal stress direction, deviation angle > 60°; the minimum principal stress direction, deviation angle > 75°) by the D-P failure criterion.

The relationship between $S_h$ and the collapse pressure gradient by the two criteria is shown in Figure 7. The collapse pressure gradient increased with decreasing $S_h$. When the saturation decreases from 60% to 15%, the collapse pressure gradient increases by

7–10%. Li et al. (2020) and Sun et al. (2018) obtained the same discovery from the analysis of stratum mechanical properties [17,20]. When the $S_h$ decreases, the strength of HBS decreases, the plastic zone increases, and the wellbore stability becomes worse. Therefore, considering the influence of the hydrate decomposition on wellbore stability, the equivalent density of drilling fluid should be appropriately increased to meet the needs of the wellbore stability after hydrate decomposition.

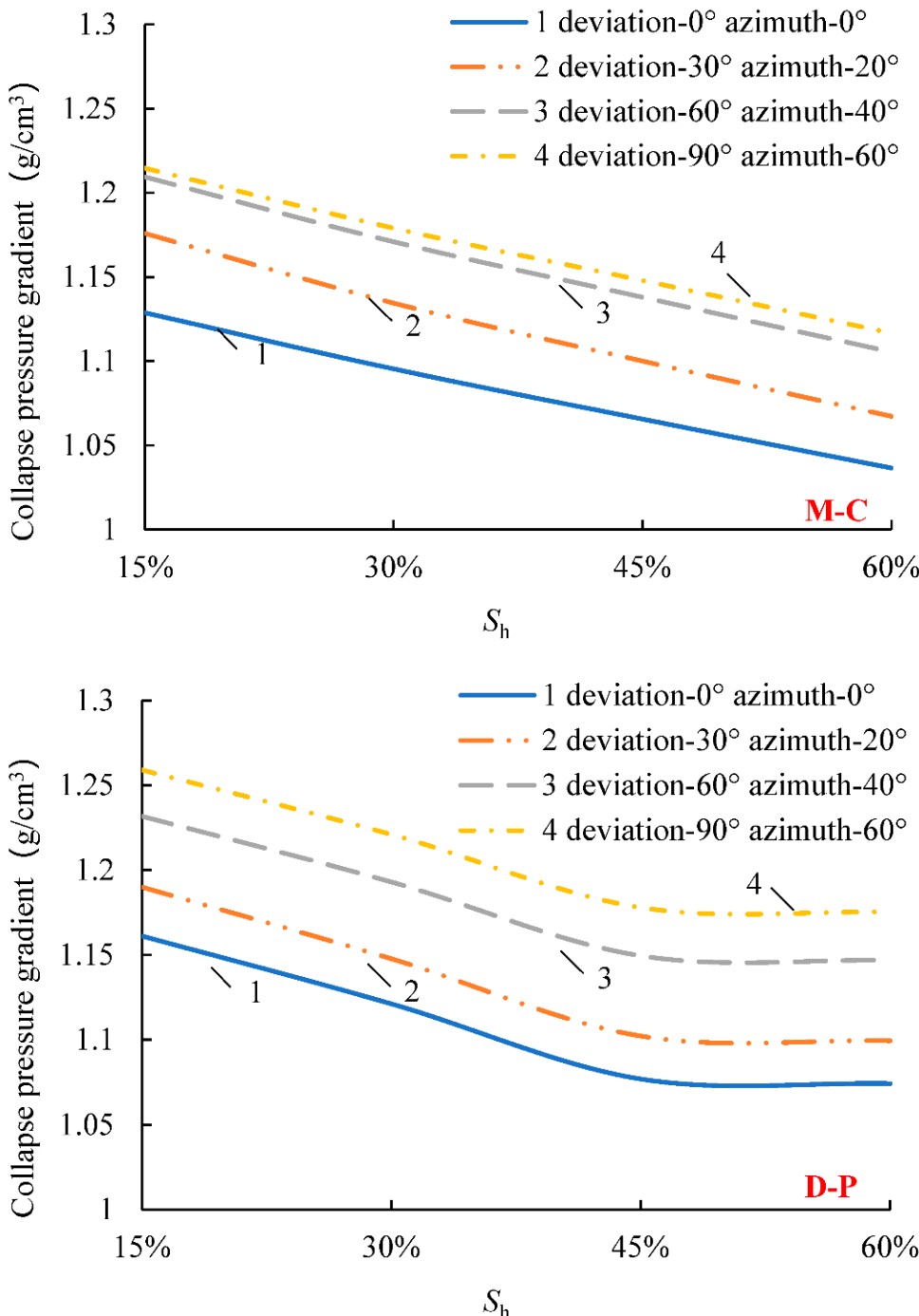

**Figure 7.** Hydrate saturation and collapse pressure gradient relationship curves by the two criteria.

By comparing the curves of the two criteria, the collapse pressure gradient decreases linearly with an increase in $S_h$ under low saturation (<45%) by the two criteria. The reason is that with decreasing $S_h$, the formation pore–space increases, and the cementation ability of hydrate to the formation weakens, which results in a continuous increase in

the collapse pressure gradient and the deterioration of the stability of the wellbore. At high saturation (>45%), the M-C failure criterion curve trend is the same as is seen at low saturation, while the D-P failure criterion curve is flat. At the same position, the calculated value of the D-P failure criterion is greater than that of the M-C failure criterion at low saturation, and the opposite is true at high saturation. It showed that the formation stability in high saturation is higher when the influence of the intermediate principal stress is fully considered. However, higher drilling fluid densities are required to maintain formation stability in low saturations.

### 4.2. Influence of the Deviation Angle on the Collapse Pressure Gradient of the Wellbore

According to the formation basic parameters and the hydrate mechanics parameters, the influence of the deviation angle on the collapse pressure gradient was calculated. Taking 15% saturation as an example, the relationship curves between the deviation angle and the collapse pressure gradient by the two criteria were obtained (Figure 8). Within a quarter of the cycle (azimuth 0~90°), the trends of the two curves are the same, and the collapse pressure gradient rises, on the whole, with increasing deviation angle. When the azimuth is less than 60° (curves 1–3), the collapse pressure gradient continues to increase with increasing deviation angle. When the azimuth is higher than 60 ° (curves 4 and 5), the collapse pressure gradient decreases first and then increases with changing deviation angle, and the position of the inflection point is gradually delayed with increasing azimuth. The collapse pressure gradient increases by 7.2–9.2% from the vertical wellbore section (deviation angle 0°) to the horizontal wellbore section (deviation angle 90°). In other words, the safe density that satisfies the horizontal section can also ensure the wellbore stability of other sections, and will not collapse.

By comparing the two curves obtained from the two criteria, it is found that when the azimuth is higher than 60°, the inflection point of the collapse pressure gradient by the M-C failure criterion appears earlier than that of the D-P failure criterion. When the azimuth is 60° and 80°, the inflection points of the M-C failure criterion curve are 10° and 20°, while considering that the inflection point of the D-P failure criterion curve, under the influence of the intermediate principal stress, is 15° and 30°. The M-C failure criterion curve increases abnormally near 65°, while the D-P curve is smooth and without distortion. The reason for this is that the yield surface of the D-P failure criterion is smooth and without sharp corners, while the yield surface of the M-C failure criterion is hexagonal and with sharp corners (Figure 4). Hence, the D-P criterion is more suitable for the wellbore stability calculation.

### 4.3. Effect of the Azimuth on the Collapse Pressure Gradient of the Wellbore

According to the formation of the basic parameters and the hydrate mechanics parameters, the influence of the azimuth on the collapse pressure gradient was calculated. Taking 15% $S_h$ as an example, the relationship between the azimuth and the collapse pressure gradient by the two criteria was obtained (see Figure 9). It was found that the collapse pressure gradient decreases and then increases with an increasing azimuth on the high inclination well sections (curves 2–6). The collapse pressure gradient reaches a minimum value in the direction of the minimum principal stress (azimuth of 90°). The same was reported in the literature [40]. With an increase in the deviation angle, the fluctuation of the curve is lower. That is, the difference in the collapse pressure gradient is not significant during drilling in any direction. When the deviation angle is 30° and 90°, the differences are 0.038 and 0.018 g/cm$^3$, respectively. In the near-vertical sections (curve 1), the minimum collapse pressure gradient was obtained at an azimuth of 60°. In summary, the wellbore should be arranged along the azimuth between 60° and 120° for high wellbore stability.

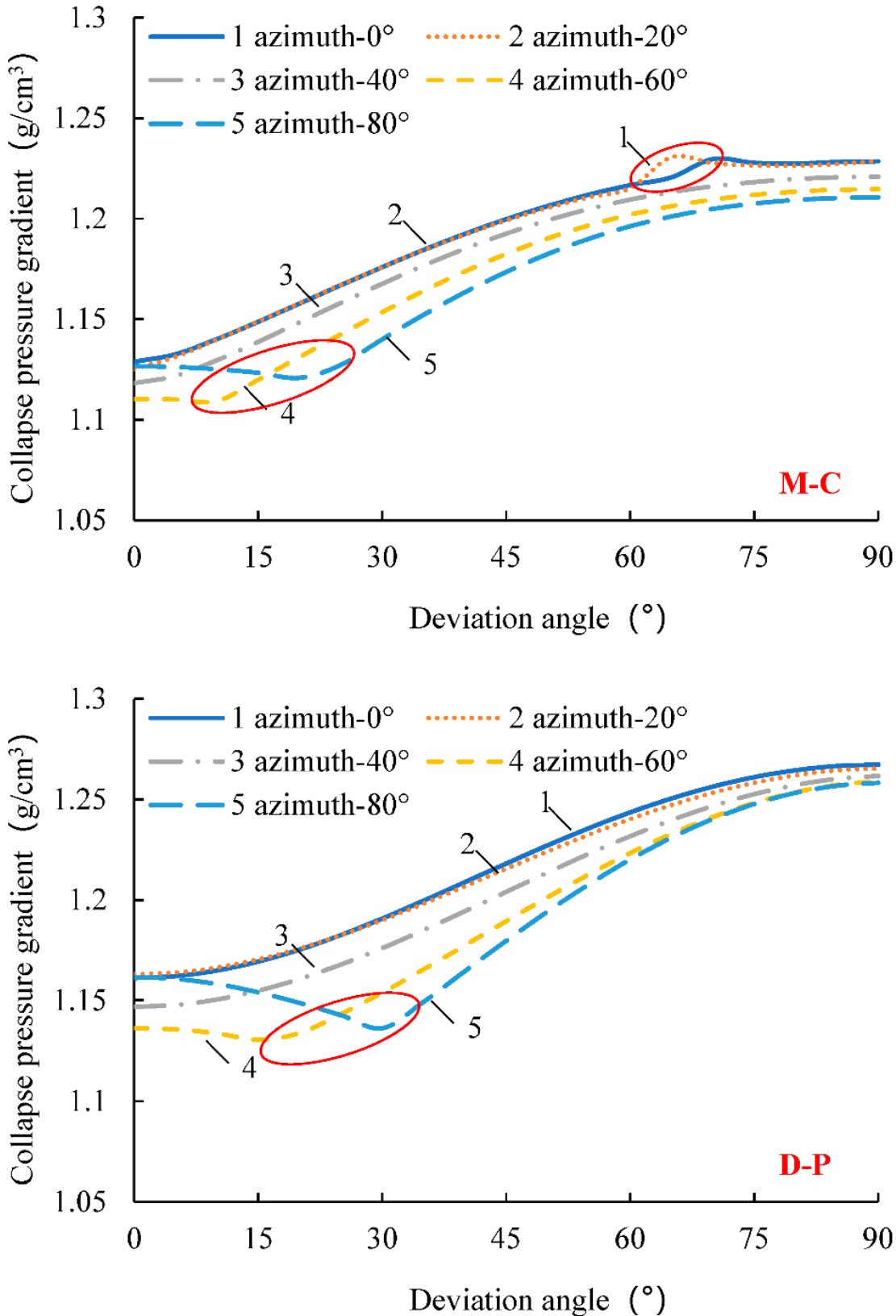

**Figure 8.** The deviation angle and the collapse pressure gradient relationship curves by the two criteria.

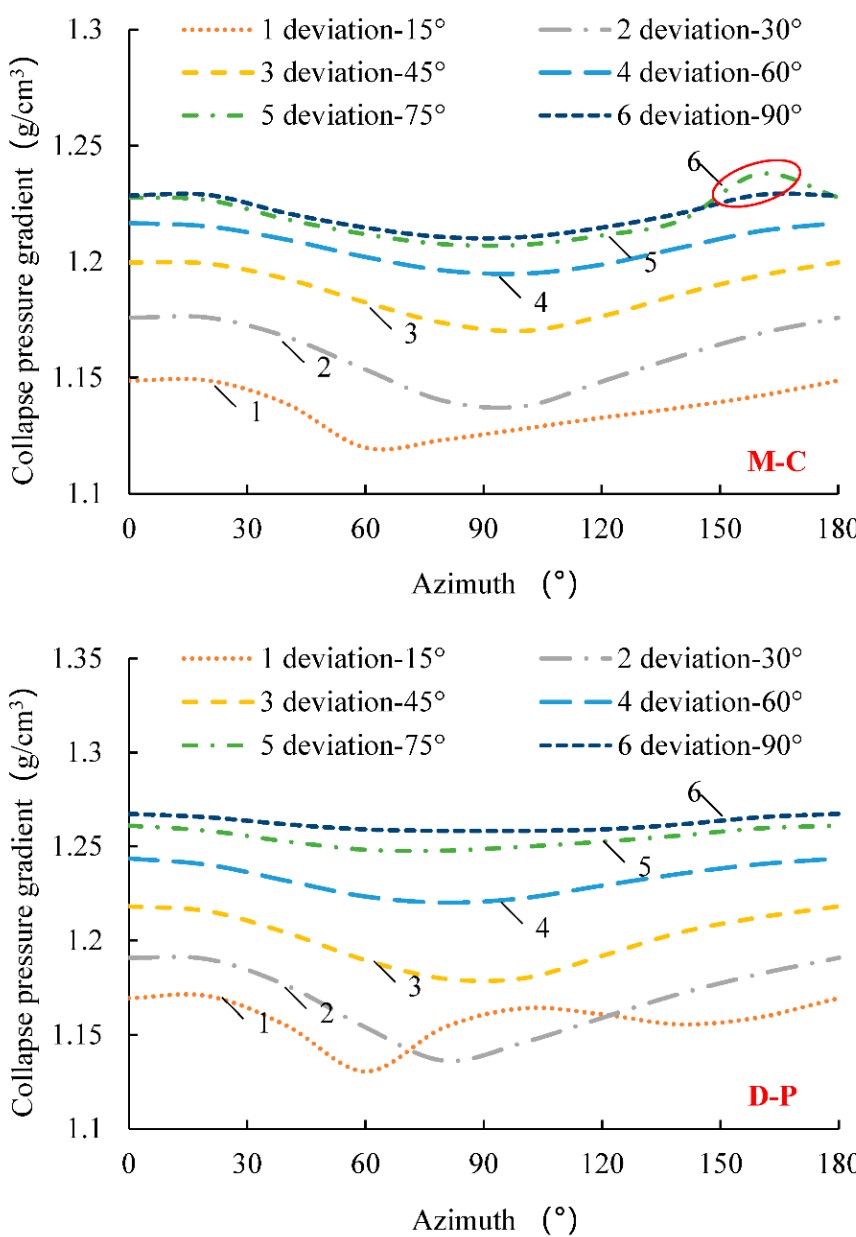

**Figure 9.** The azimuth and the collapse pressure gradient relationship curves by the two criteria.

*4.4. Fracture Pressure Gradient and the Safe Drilling Fluid Density Window*

The safe drilling fluid density window is defined as the difference between the collapse pressure gradient and the fracture pressure gradient. It is safer for drilling with a relatively wide safe drilling fluid density window. Therefore, the variation pattern of fracture pressure was analyzed first. According to the formation of the basic parameters and the hydrate mechanics parameters, and taking the 15% $S_h$ as an example, the relationship curve between the azimuth and fracture pressure gradient was simulated, as shown in Figure 10. When the deviation angle was higher than 45° (curves 4–7), the fracture pressure gradient increased and then decreased with increasing azimuth, and the maximum value is in the direction of the minimum principal stress. When the deviation angle is less than 45° (curves 1–3), the fracture pressure gradient generally increases and then decreases with increasing azimuth, but the peak value exists between 120° and 150°. The azimuth corresponding to the peak decreases gradually with increasing deviation angle, closer to the direction of the minimum principal stress. During the drilling of horizontal wellbores, the drilling fluid density needs to meet the stability of the wellbore in all sections. Therefore, a drilling fluid density is

selected in the yellow zone formed below the vertical and horizontal wellbore curves (curves 1 and 7), which can be safely drilled in any direction.

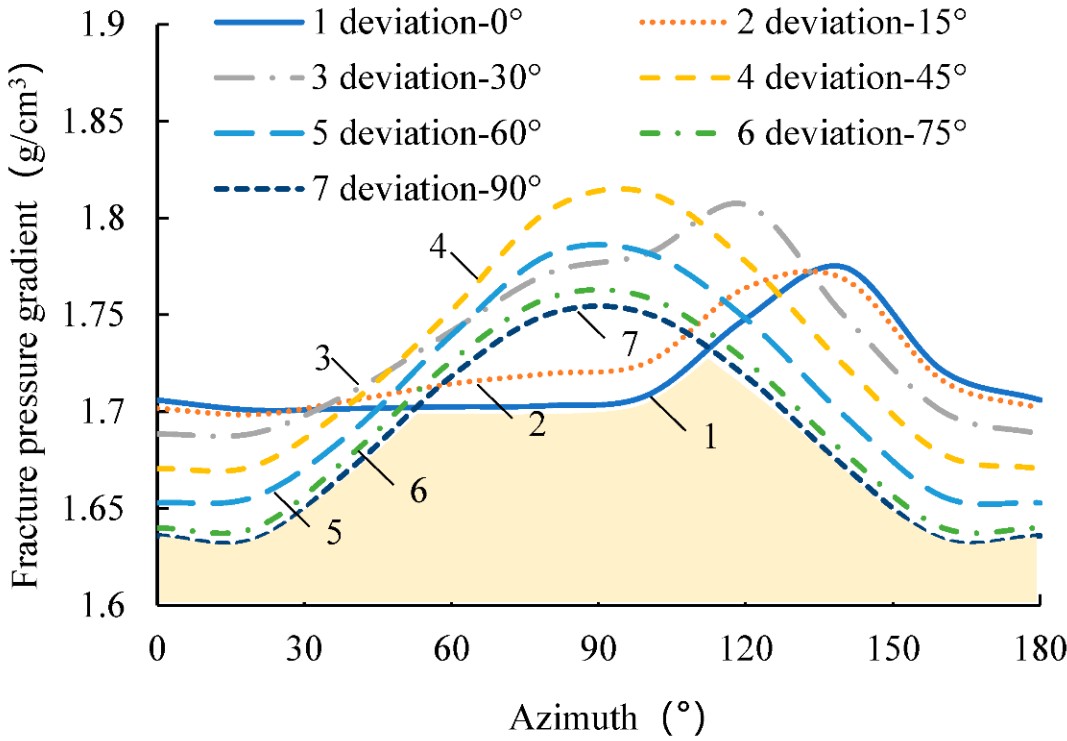

**Figure 10.** The azimuth and the fracture pressure gradient curves of the wellbore.

The cloud diagram of the safe density window is drawn by the lower hemisphere projection method, as shown in Figure 11. The safe density window has a center-symmetric strip distribution in the range of 0.368–0.660 g/cm$^3$. The density window of the upper and lower belt regions is narrow, while the density window of the middle region is large. The maximum safe density window is the "crescent" region in the direction of the minimum principal stress. Drilling operations near this zone are most conducive to wellbore stability.

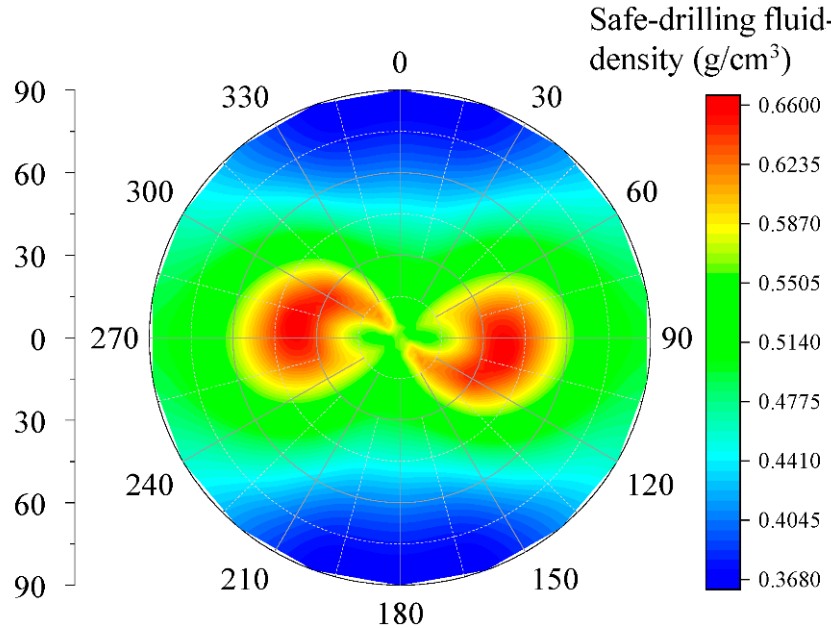

**Figure 11.** Window distribution cloud diagram for safe drilling fluid density.



## 5. Conclusions and Suggestions

1. The window distribution cloud chart of the collapse pressure gradient and the safe drilling fluid density of HBS-CS has a centrosymmetric distribution with a deviation angle and azimuth.

2. Hydrate decomposition will lead to a higher collapse pressure and poorer stability of the formation. Therefore, the drilling fluid density should be appropriately increased by 7~10% during drilling to ensure the stability of formation after hydrate decomposition.

3. The collapse pressure gradient increases by 7.2–9.2% from the vertical wellbore to the horizontal wellbore. From the perspective of preventing wellbore collapse, the safe density that satisfies the horizontal section can also ensure the wellbore stability of other sections. To prevent the wellbore fracture, a safe density that satisfies both the horizontal and vertical sections is necessary to ensure wellbore stability in all sections.

4. Considering the combined effects of collapse, fracture pressure gradient, and safety density window, it is suggested that the borehole be arranged along the azimuth of 60–120°, which could greatly reduce the risk of the drilling operation.

The model can provide a safe density window in horizontal drilling design and guide the selection of drilling fluid for construction. In case of an emergency, such as a kick and overflow, the risk of wellbore instability can be calculated quickly by this model to guide the formulation of corresponding disposal measures. However, the parameters involved in the model are not comprehensive enough, and the calculation accuracy is slightly insufficient. Therefore, the elastic model is just the beginning of the HBS-CS wellbore stability study. Next, we will establish the plastic model and damage model and strive to improve the prediction accuracy of wellbore stability, helping NGH mining.

**Author Contributions:** Conceptualization, Q.H.; data curation, Y.L. and Y.Z.; methodology, X.S. and Y.L.; software, M.C. and Y.Z.; validation, M.C.; writing—original draft, X.S. and Q.H.; writing—review and editing, Q.H. All authors have read and agreed to the published version of the manuscript.

**Funding:** This study was supported by the Key R&D Projects of Hainan Province (Grant no. ZDYF2022SHFZ063) and the Shandong Special Fund of Pilot National Laboratory for Marine Science and Technology (Qingdao) (Grant no. 2021QNLM020002).

**Institutional Review Board Statement:** Not applicable.

**Informed Consent Statement:** Not applicable.

**Data Availability Statement:** Not applicable.

**Conflicts of Interest:** The authors declare no conflict of interest.

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
