# Peer review of "Stability Characteristics of Horizontal Wells in the Exploitation of Hydrate-Bearing Clayey-Silt Sediments"

_jmse, doi:10.3390/jmse10121935_

Round 1
Reviewer 1 Report
Dear authors of the article. Thank you for an interesting and relevant research with growing potential today. The construction technology of directional and horizontal wells, complicated by geological heterogeneity, is complex and difficult to predict at the stage of field work. Therefore, this study certainly has further development at the stage of field testing and the identification of empirical coefficients to refine the above calculation.
There are a number of small comments and recommendations on your article:
1. I propose to change the title of the article, since the original title is not the essence of the work on the development of a well profile depending on development.
2. In line 13, the text is formatted by the author, namely mail.
3. In annotations it is not recommended to enter abbreviations and indicate abbreviations of abbreviations in brackets.
4. Line 28 «Keywords» - you need to add 2 more additional words.
5. In line 47 there is a sharp transition to the width. I advise you to make a smoother transition to the task and start the sentence with a new paragraph.
6. In line 60, it is recommended to start the sentence with a paragraph.
7. Line 66 is missing points of view at the end of the sentence.
8. In line 83, you should justify why such pressure values ​​​​are chosen.
9. Line 90. Table of contents. Write MPa per MPa in columns 3 and 4 respectively.
10. Section 2 ends in a table, which is incorrect in terms of collection. You must complete a section in order to move on to the next section.
11. Line 102, Figure 2 of reduced quality, the quality of the drawing should be improved.
12. Line 204 has an extra dot before the Equation.
13. In line 206 there is no description for the coefficient K. And what does «cot» mean? If it is a cotangent, then it should be denoted as «ctg».
14. Line 213, Figure 4 of reduced quality, the quality of the Figure should be improved.
15. Everywhere it is better to remove the words «Equation» and write (ordinal number of the equation).
16. The MDPI design format provides for the design of references to sources in square brackets with a ordinal number as they are mentioned in the article (look at the design template again).
17. In table 4 (line 243), the dimension corresponds to the parameters after the comma, and not in brackets.
18. No descriptions of solutions were found that fit the approximation, within acceptable limits.
19. There is no comparison of the author's study with similar studies of other scientists, both in section 1 and in Discussions (which also does not occur as separate sections), starting from the introduced simplified models.
20. The Conclusions section should deliberately end the continuation of the studies and plans for further work at the end and introduce an introductory sentence before the dot numbering.
21. Not enough sections «Financing» instead of «ACKNOWLEDGMENTS»; «Statement of the Institutional Review Board»; «Statement of Informed Consent»; «Data Availability Statement»; «Conflict of interests».
22. The «References» section must be designed in accordance with the rules for the design of the MDPI. It is worth noting that there are no DOI or access links among the sources.
Reviewer 2 Report
In this manuscript, the authors presented their study for mechanical properties of hydrate-breaking strata in the clayey-silt sediments. Their analysis was based on Mohr-Coulomb criterion and Drucker-Prager criterion and elastic theory. I have the following comments for the authors considering in the revision.
(1) The usage of Mohr-Coulomb criterion and Drucker-Prager criterion and elastic theory require justification.
(2) In Section 2, the authors used the technical properties of a specific site in Shenhu area, northern South China sea. It is better to include a local map to show where the site is located.
(3) How did you determine the hydrate saturation. It seems quite low. Does it mean the seabed in unsaturated soil? (Table 1). Also, the cohesion is quite low.
(4) It seems that no any validation of the proposed theoretical work.
(5) It is necessary to clearly state the limitation of the proposed model? Also, how the readers applied the proposed work to engineering design?
